# Gender Differences in Anthropometric, Functional Capacity Measures and Quality of Life in Individuals with Intellectual and Developmental Disabilities

**DOI:** 10.3390/jfmk9020084

**Published:** 2024-05-05

**Authors:** Miguel Jacinto, Diogo Monteiro, Filipe Rodrigues, Susana Diz, Rui Matos, Nuno Amaro, Raul Antunes

**Affiliations:** 1ESECS, Polytechnique of Leiria, 2411-901 Leiria, Portugal; miguel.s.jacinto@ipleiria.pt (M.J.); filipe.rodrigues@ipleiria.pt (F.R.); sucris.diz@gmail.com (S.D.); rui.matos@ipleiria.pt (R.M.); nuno.amaro@ipleiria.pt (N.A.); raul.antunes@ipleiria.pt (R.A.); 2CIDESD, Research Center in Sport, Health, and Human Development, 5001-801 Vila Real, Portugal

**Keywords:** anthropometry, body composition, disability, functional capacity, strength, quality of life

## Abstract

The aim of the of the current investigation was to investigate the possible differences concerning males and females in anthropometry, body composition, functional capacity, strength and quality of life variables. After obtaining signed informed consent, 37 participants (18 males; 19 females), with mean age of 39.08 and standard deviation of 11.66 years, voluntarily participated in this study. Anthropometry, body composition, functional capacity, strength, and quality of life were assessed using validated and reliable instruments and tests for this population. The males and females were compared using a Mann–Whitney U signed rank test. Significant differences were detected among the following variables, height (*p* = 0.028), body mass index (*p* = 0.033), fat mass (*p* = 0.002), muscle mass (*p* ≤ 0.001), phase angle (*p* = 0.005), medicine ball throwing strength (*p* = 0.010), and peak toque left knee (*p* = 0.028), with males showing better results in all the variables. The sample in this study showed differences in the anthropometric, composition, and strength variables. Studying this population can help ensure that everyone has equal access to services and adequate support for their personal needs, improving their quality of life.

## 1. Introduction

Anthropometry and body composition assessments are non-invasive, economical, and widely employed in evaluating the changes in weight and body dimensions as individuals age [1]. The literature seems to indicate that males and females differ in terms of anthropometric and body composition markers [2,3,4,5], and these differences have important implications for health and well-being, particularly with the onset of cardiovascular and metabolic diseases, as well as in the assessment and prescription of exercise programs and sex-specific health interventions. These differences relate to the physical disparities between males and females in terms of body measurements and proportions (males show greater shoulder length, hand and foot sizes, neck length, and more absolute muscle mass and less fat mass than females), which could be due to hormonal differences, such as greater testosterone production in males [6].

Limitations in functional capacity reduce people’s quality of life due to a lack of independence, higher healthcare expenditure, and increased mortality risks [7,8]. Although the average life expectancy is higher among females, they report more functional limitations and physical disabilities [9,10,11]. The greater capacity for strength production in males is correlated with a higher ratio of type II muscle fibers, which are known to be particularly effective at producing strength and power [12].

The ability to produce strength has a significant role to play in sustaining body weight and supporting, assisting, and absorbing the impact of activities, and is therefore a factor that predicts people’s functional ability and quality of life [13,14,15]. Other studies show that the sex differences in quality of life can depend on various factors, including social, economic, cultural, and biological aspects [16,17,18].

Although there are substantial studies in the literature describing the sex differences in anthropometry, body composition, functional capacity, strength, and quality of life [9,10,11,19], studies on the population with Intellectual and Developmental Disabilities (IDDs) are uncommon, or have a minor number of participants, which limits the comprehension of sex-related differences in this population as well as the creation of differentiated promotion strategies to promote the quality of life of males and females with an IDD [20,21,22,23,24]. Due to them having a sedentary and inactive lifestyle, it is clear that people with an IDD report high body mass index scores and a poor physical condition, including strength-related concerns [25,26]. These reduced values of physical ability variables are correlated with a decrease in their physical and functional capacities (decreasing their functional ability and achievement in carrying out the activities of daily life) [27], health, and life expectancy [28]. They are also associated with cardiovascular risk, indicating that elevated strength correlated with a reduction in waist circumference and an increase in triglycerides in children, adolescents, and adults [29].

Indeed, the elevated levels of upper and lower muscle strength observed in adults are linked to a decreased risk of mortality [30].

Considering the biological, social, and cultural characteristics of the population without disabilities who may suffer from different kinds of healthcare problems, this study on the population with IDDs can help ensure that everyone has equal access to services and adequate support for their personal needs. It can also help adjust the offering of intervention programs or the prescription itself based on the specific characteristics and needs of each sex, enhancing their quality of life. The aim of this study was to examine the differences between the males and females in anthropometric, body composition, functional capacity, strength, and quality of life variables. We hypothesized the following: (a) the females had higher anthropometric scores; (b) the females had higher body composition scores; (c) the females had lower functional capacity scores; (d) the females had lower strength scores; and (e) the females had lower quality-of-life scores.

## 2. Materials and Methods

### 2.1. Research Design

This work follows a cross-sectional design in agreement with the fundamentals of the Declaration of Helsinki [31]. Ethical authorization to investigate Sport and Exercise Sciences was obtained from the Ethics Committee of the Faculty of Sport Sciences and Physical Education at the University of Coimbra under the code CE/FCDEF-UC/00872021. Subjects were recruited for this study from an institution that provides services to people with disabilities. Prior to participation, all the individuals and their families obtained full knowledge about the aim and procedures of this study. Informed consent ensued obtained from all of the participants and their guardians through signed consent forms.

### 2.2. Participants

This study complied with people who met the following inclusion criteria: (1) adults with an IDD; (2) no medical contraindications; (3) the ability to perform movements such as pushing or pulling; and (4) the ability to complete the planned assessments. At the same time, this study included people who met the following exclusion criteria: (1) dependence on walking supports for mobility; (2) a profound IDD; (3) communication difficulties; and (4) the failure to present a signed informed consent form. Our study included 37 participants institutionalized in a support institution, with a mean age of 39.08 years (SD = 11.66), varying from 20 to 58 years. From all the participants, 49% (*n* = 18) were male, and 51% (*n* = 19) were female. The participants’ family members/carers/caregivers also took part in this study.

### 2.3. Instruments/Procedures

#### 2.3.1. Anthropometry

A stadiometer scale, specifically model 870, was utilized to evaluate body weight and height. The participants were instructed to stand barefooted on the platform, touching the pole of the stadiometer, maintaining a conventional positioning their arms alongside their body. Their BMI was subsequently find out through weight (kilograms) divided by height (meters squared). These measurement techniques were found to be feasible, reliable, and accurate for individuals with an IDD, as supported by prior studies [32,33,34,35].

#### 2.3.2. Body Composition

To assess body composition, including fat and muscle mass, as well as phase angle, we used bio-impedance equipment (InBody770, Cerritos, CA, USA). The participants stood barefoot on the device platform, ensuring contact with the four electrodes on their feet. Weight was measured, and the participants held onto a bar that has been provided with four electrodes for their hands during the assessment. This method is widely recognized for its feasibility, reliability, and non-invasive nature [36].

#### 2.3.3. Functional Capacity

Several tests from the Senior Fitness Tests [37,38] were employed to evaluate physical fitness, specifically, the 30 s sit-to-stand test, which assesses the endurance and strength of the lower limbs by measuring the total number of repetitions performed in 30 s. The trial began with the participants seated centrally in a chair, maintaining straight back, with their feet at shoulder level and resting on the ground. Upon being given the “start” signal, the participants stood up to full extension (upright position), and then proceeded to take the original seated position. This test has been validated for the IDD population [34,39]. The “Timed up and Go” test assessing physical mobility, focusing on speed, agility, and dynamic balance, was also used. The participants began by sitting on a chair, with their hands resting on their legs and their feet surface on the ground. Upon being given the start signal, they rose from the chair and walked as fast as feasible across a pinecone positioned 2.44 m away, and then proceeded to the chair. The participants were informed that their performance in the test was measured by the time taken to finished the test and has been validated for the population with IDDs [39]. Last, the 6 min walk test assessed aerobic endurance. The participants were instructed to start walking as fast as viable (without running) along a space marked by cones after the start signal. If required, the participants could take a break and rest, with the possibility of sitting down, before resuming the course, which is validated and reliable for the population with IDDs [40].

#### 2.3.4. Neuromuscular Capacity

The “3 kg medicine ball throw test” was performed to determine upper limb strength. The participants started seated on a chair holding a medicine ball positioned next to their chest. Initially, they threw the ball with the aim of reaching as far as possible from the chest. This test has been validated for individuals with IDDs according to several authors [34,41].

Lower limb strength was evaluated using an isokinetic dynamometer device (model: BIODEX Multijoint System 3 Pro, locate Shirley, NY, USA). This evaluation engaged maximal concentric contractions for flexion and extension of the leg. Prior to the assessment session, the equipment was adjusted according to the producer’s guidelines. To explain the test, 3 repeats were executed for every action beforehand beginning the test [42]. Through the assessment, the individuals’ were instructed to cross their arms with their hands on the opposite shoulder. The warm-up protocol consisted of 5 min of walking at a moderate intensity. Concentric muscle contractions were evaluated, with 3 repetitions of the movement at 60º/s. An interval of 60 s has been determined between the familiarization repetitions, tests, and the different angular velocities [43]. The peak torque values were obtained. This test has been shown to be reliable for the target population [44].

#### 2.3.5. Quality of Life

The 48-item Personal Outcomes Scale Portuguese version [45] was administered by technicians who had received specialized training for this purpose. It is composed for factors (independence, Social Participation and well-being) and domains (Personal Development; Self-determination; Interpersonal relations; Social inclusion; Rights; Emotional, Physical, and Material well-being) The scale was answered by individuals with IDDs (self-report) and by their caregivers (third-party report). The measure comprises six dimensions and were aggregated using the mean of each dimension. The responses were assessed using a 3-point Likert scale, where higher results show a better quality of life (e.g., 3 = always; 2 = sometimes; 1 = rarely or never). Regarding the self-report assessment, the composite reliability coefficients for each dimension were Personal Development = 0.65; Self-determination = 0.73; Interpersonal relations = 0.48; Social inclusion = 0.73; Rights = 0.77; Emotional well-being = 0.46; Physical well-being = 0.46; Material well-being = 0.74. Likewise, for the third-party measure, the scores were Personal Development = 0.78; Self-determination = 0.76; Interpersonal relations = 0.61; Social inclusion = 0.80; Rights = 0.79; Emotional well-being = 0.73; Physical well-being = 0.55; Material well-being = 0.70.

### 2.4. Statistics Analysis

The mean and standard deviation were estimated for the variables below in this study. The normality of the data was verified using the Shapiro–Wilk (*n* < 50) test, and homoscedasticity was analyzed using the Levene test, as suggested by Ho [46]. The results showed that the sample does not exhibit a normal distribution for the variables under study (*p* < 0.05). In terms of homoscedasticity, the results showed that the variances are equal between groups (*p* > 0.05) as suggested by Ho [46]. Therefore, the Mann–Whitney U test was used to compare differences across studied variables. The significance level for rejecting the null hypothesis was set at 5%, and analyses were carried out using IBM SPSS version 29.

## 3. Results

Table 1 shows the descriptive statistics and the differences between the males and females in terms of the anthropometric and body composition variables. The results show significant sex differences in BMI (*p* = 0.033). There are also significant differences between the males and females in the other body composition variables. There are no significative differences in the three functional tests performed (*p* ≥ 0.05). Regarding the neuromuscular capacity tests, we found that the males showed higher values than the females, although they were only significant in the ball-throwing test (*p* = 0.010) and the peak torque of left-leg extension (*p* = 0.028). Last, no differences between the males and females in terms of the quality-of-life responses were found (*p* ≥ 0.05).

## 4. Discussion

The aim of the current investigation is to verify the sex differences in anthropometry, body composition, functional capacity, strength, and quality of life in individuals with IDDs. The results indicate that there are significant differences in height, thus partially confirming Hypothesis (a). Similarly, there existed significant differences in all the body composition variables, verifying Hypothesis (b). The results do not confirm Hypothesis (c), even though the males presented greater values. Hypothesis (d) is partially confirmed, as no significant differences were found in all the variables assessed in terms of neuromuscular capacity. Finally, hypothesis (e) was not confirmed, as there were no significant differences in the different quality-of-life responses.

The results of our study for the anthropometry and body composition variables are in line with the results for the population without disabilities, showing that the females have poorer values than the males [2,3,4,5]. In the same sense, our results seem to verify the previous studies results in the population with IDDs [21,47,48,49,50], in the sense that there are significant sex differences, and even when these differences are not significant, the female showed poorer values. Females with IDDs also show lower values when compared with females without disabilities [51,52]. In addition, the participants in the current study displayed scores indicating them having overweight and/or obesity [53], consistent with previous studies, which increases the possibility of obesity-related health problems, such as cardiovascular or metabolic diseases [48,54] and premature death [55,56]. This fact can be induced by various factors, including environment, behavioral, genetic (such as Bardet–Biedl syndrome or Prader–Willi syndrome), and/or medical factors, in particular, neurological and/or metabolic disorders [50,57], namely, limited access to health care, eating unhealthy foods [58,59], and taking several prescribed medications [60], which are variables that should be assessed and controlled in future studies.

In the population without disabilities, the literature is clear and indicates that there are significant differences in functional capacity [9,10,11]. Still, our results did not show significant differences; the females also presented poorer values, corroborating the previous studies on this population [20,21,61]. For example [20], when analyzing Special Olympics athletes, the authors found that the male participants had higher scores than the female athletes in the timed sit-to-stand test. In a cross-sectional analysis, another group [21] identified higher values in males compared to females for the 30 s chair test, timed up-and-go test, and 6 min walk test, indicating that could be some variability among adults with IDDs, and thus future studies are needed to create normative values and examine the reasons for these possible differences.

The males in this study always showed higher values in all the neuromuscular capacity tests carried out, supporting studies on the population without disabilities [9,10,11]. For several of the variables assessed, some of them are even significant. The previous studies on the population with IDDs also indicate that females have lower muscle strength compared to males [20,22,62]. Cuesta-Vargas and collaborators [20] observed that the male participants obtained higher scores than the female athletes in several neuromuscular capacity tests. Also Lahtinen and collaborators [22] found that males consistently showed greater abdominal strength and endurance than females. More recent documentation has shown comparable sex-based disparities in strength within a cohort of Nordic Special Olympics athletes [23]. Low levels of strength could be caused by the peripheral and central activation of motor units and some abnormal intrinsic muscle characteristics in the IDD population [63,64].

Although studies on the population without disabilities show significant differences between males and females in terms of quality of life [16,17,18], our results do not confirm this information. Although research into these variables is scarce, the current studies on the population with IDDs seem to indicate that there are no significant differences in perceived quality of life between males and females [21,65].

The findings of the present exploratory study expand on the current evidence, highlighting the importance of constant anthropometric and body composition assessments associated with risk values. Likewise, it is essential that health professionals explore potential alternative medications that do not induce weight gain, as well as other behavioral strategies that can serve as substitutes or decrease their impact, such as an adjusted diet. This has to be taken into account, especially in females, as sex is also associated with weight gain [48,51]. This study provided important insights into the different characteristics and needs of males and females with IDDs. Most studies are carried out in the same way for both sexes, which may not be appropriate considering the basic differences between males and females. Prescribing physical activity, exercise, or sport equally for both sexes seems to make little sense. While it is important to analyze and compare men and women on these variables to have more information to prescribe and adapt the offer, it is also crucial to consider these differences when organizing intervention programs and try to understand the role that exercise/sport can play for both.

In addition to these strategies, it is also essential that physical activity should be part of the lives of individuals with IDDs [66], as responses to this practice also play an important role in promoting and/or maintain functional capacity and physical fitness, promoting the execution of activities of daily live and reducing the risk of early mortality in adults with IDDs [67,68]. Similarly, physical activity can act as a predictor of improved quality of life (β11 = 0.703, *p* < 0.001) [69]; therefore, the development of strategies and tools to maintain/adopt healthy and active lifestyles should include regular physical activity [70,71]. Additionally, the findings from research conducted by Tomaszewski and colleagues [72] revealed a significant association between the total quality-of-life scores and weekly step count. Their study indicated that for every additional 1000 steps for every day, the total quality of life result increased by 2.56 points. Also, in Pérez-Cruzado and Cuesta-Vargas’ study [73], a physical activity intervention program with an 8-week of duration improved the physical ability and quality of life of 40 individuals with IDDs.

If, on the one hand, this regular routine of physical activity is conditioned by the existence of barriers (namely, the lack of adapted programs, high financial cost, or lack of motivation) [74], some more recent approaches have proposed strategies to mitigate/reduce them. In the study by Ferreira et al. [75], two programs are presented that have been adapted, one of which uses low-cost materials. These programs have already been shown to be effective and to have an acceptable adherence rate [76]. Tomé et al. [77] also presents an intervention program using play, games, and pre-sport modalities, increasing enjoyment and the motivation to practice.

Despite expanding the current evidence, some limitations in this study may limit the analysis and generality of the results, which should be reported. Firstly, this study did not include information on the etiology of the participants, namely the supports needed, medication, or other comorbidities, and future studies should examine the possible interactions between these factors or potential explanations [78]. Other variables may also play a part in this equation, such as the amount of physical activity practiced and/or diet. The small sample size is also a limitation, although it confirms the results of previous studies. The absence of knowledge on the existence of syndromes in the participants may also affect the interpretation of the results of this study [78]. The application of a randomized intervention study could also better explain the relationship between the variables studied and whether the groups assimilated post intervention. Further research is needed to identify additional factors associated with anthropometry, body composition, functional capacity, strength, and quality of life in individuals with IDDs, not only for a better understanding of the unfavorable values, but also sex differences.

## 5. Conclusions

The participants in this study appear to suggest significant differences in some anthropometric, body composition, and strength variables, with the females exhibiting poorer values. The implications of these results for policy and exercise are significant and emphasize the significance of a constant focus on evaluation and examination, and the promotion of physical activity and exercise within the population with IDDs.

## Figures and Tables

**Table 1 jfmk-09-00084-t001:** Descriptive statistics and sex differences in anthropometric and body composition measures.

	Total (*n* = 37)	Males (*n* = 18)	Females (*n* = 19)	Mann–Whitney U	Sex Differences
Measures	M ± SD	M ± SD	M ± SD		
Age (years)	39.08 ± 11.66	41.16 ± 9.62	37.10 ± 13.27	−1.049	-
Height (cm)	159.77 ± 8.51	163.47 ± 6.87	156.26 ± 8.58	−2.203	*p* = 0.028
Weight (kg)	73.38 ± 17.11	69.55 ± 14.46	76.64 ± 18.82	0.403	-
BMI kg/m^2^	28.9 ± 6.9	26.07 ± 4.13	31.32 ± 7.91	−2.127	*p* = 0.033
Fat mass (kg)	26.68 ± 14.08	18.73 ± 8.94	33.44 ± 14.28	−3.130	*p* = 0.002
Muscle mass (kg)	25.87 ± 5.41	28.51 ± 5.29	23.62 ± 4.51	−3.328	*p* ≤ 0.001
Phase angle (º)	5.75 ± 1.01	6.15 ± 0.84	5.41 ± 1.01	−2.799	*p* = 0.005
Sit to stand (repetitions)	13.08 ± 3.35	13.55 ± 4.20	12.63 ± 2.31	−0.780	-
Timed Up and Go (s)	8.50 ± 3.26	8.40 ± 3.49	8.59 ± 3.12	−0.593	-
6 min walk (m)	476.62 ± 108.73	486.32 ± 138.32	467.44 ± 73.92	−1.444	-
Medicine ball throw test (m)	2.36 ± 0.67	2.59 ± 0.73	2.14 ± 0.55	−1.597	*p* = 0.010
PT cc KE 60º/s right (N.M)	71.05 ± 38.44	84 ± 44.41	58.79 ± 27.66	−1.808	-
PT cc KE 60º/s left (N.M)	72.02 ± 37.59	86.29 ± 45.33	58.50 ± 22.10	−2.203	*p* = 0.028
PT cc KF 60º/s right (N.M)	41.38 ± 27.09	49.25 ± 32.21	34.33 ± 19.85	−1.170	-
PT cc KF 60º/s left (N.M)	38.83 ± 22.80	44.28 ± 27.36	33.66 ± 16.58	−1.094	-
QoL—self report	92.43 ± 10.06	91.16 ± 10.85	93.63 ± 9.39	−0.487	-
QoL—third party report	85.91 ± 11.46	86.22 ± 3.49	85.63 ± 10.93	−0.228	-

Notes: M = mean; SD = standard deviation; *p* = significance level; PT = peak torque; cc = concentric; KE = knee extension; KF = knee flexion.

## Data Availability

All data supporting were included on this paper.

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
