# Peer review of "Gender Differences in Anthropometric, Functional Capacity Measures and Quality of Life in Individuals with Intellectual and Developmental Disabilities"

_jfmk, 2024, doi:10.3390/jfmk9020084_

Round 1

Reviewer 1 Report

Comments and Suggestions for Authors

My recommendations are the following:

Abstract- I recommend rewriting the purpose, fully mentioning the targeted aspects. The first sentence is background, therefore it cannot be assimilated with the purpose of the study.

Lines 15-16 recommend mentioning that you refer to the arithmetic mean and standard deviation of the age of the subjects.

Lines 16 – 18 recommend mentioning which test tools were used. Anthropometric measurements are not a method, as well as the evaluation of motor qualities are determined by tests and not methods. I recommend rewriting the Method section of this Abstract.

I recommend rewriting the conclusion, it is not focused.

I recommend adding the word - disability to the keywords.

Introduction- I recommend that this part be focused on the typology of the sample. In the first paragraphs, anthropometric and functional aspects are presented in a general way. I recommend that each of the aspects covered in the introduction should cover the topic of the study.

Bibliographic index 2, present in lines 29-32, refers to another category of people, does not show the age of the subjects, it is from 1990, I recommend its deletion. Anthropometric trends have changed in the last 30 years, it is a secular trend.

Subsection 2.3.5. I recommend mentioning the 6 dimensions of the questionnaire. I recommend mentioning the reliability coefficient for the entire questionnaire and for the 6 dimensions, value for each sample.

I recommend that caregivers be mentioned in the participants section as part of the experiment, or the clarification.

Table 1, I recommend that the number of subjects be mentioned on the first or second line. Also under the table you mention p = significance level, it does not appear from the table, I recommend clarification.

Section 3 does not emerge from the table and its interpretation of the 6 dimensions of the Personal Outcomes Scale Portuguese version questionnaire, I recommend clarification. Only present a total value.

I recommend that it be mentioned what is the proportion of sexes in the third party report, aspects related to numbers, gender, etc. about them were not mentioned in any section.

Taking into account the statistical indicators mentioned in section 2.4, they do not fully emerge in section 3. Results. I recommend that all parameters calculated for each indicator should be mentioned in table 1. Mention what is the significance value p.

Section 2.4. I recommend that you mention that the reliability of the questionnaire and the indicator were calculated.

Lines 213-214 recommend to mention which category of persons you refer to, this purpose is too general.

Line 303 - I recommend the correction, those with profound deficiency are excluded from the study, according to the previously mentioned criteria. I recommend that it should be mentioned in the statements that no interventional program was applied.

I recommend rewriting the Conclusions section focused on the results of this study. I sound too general.

In conclusion, he recommends that the statistics be improved, concrete and relevant correlations be made and the interpretation of the results be focused. The introduction should be majorly revised.

Author Response

Response to REVIEWER 1

Abstract- I recommend rewriting the purpose, fully mentioning the targeted aspects. The first sentence is background, therefore it cannot be assimilated with the purpose of the study.

Response: We've removed the background. The purpose has been improved according to the reviewer's suggestion. Thank you.

Lines 15-16 recommend mentioning that you refer to the arithmetic mean and standard deviation of the age of the subjects.

Response: Thank you for your suggestion, which has been accepted.

Lines 16 – 18 recommend mentioning which test tools were used. Anthropometric measurements are not a method, as well as the evaluation of motor qualities are determined by tests and not methods. I recommend rewriting the Method section of this Abstract.

Response: Dear reviewer, as you can see in the methods section, we applied various instruments/methods which, if we presented them all in the abstract, would make it difficult to read. We have redrafted the abstract, but we have chosen not to present the various instruments/methods used/applied.

I recommend rewriting the conclusion, it is not focused.

Response: We've added a sentence about the conclusion to the abstract. On the other hand, we've left the last sentence as a practical implication.

I recommend adding the word - disability to the keywords.

Response: The word "disability" has been added to the keywords.

Introduction- I recommend that this part be focused on the typology of the sample. In the first paragraphs, anthropometric and functional aspects are presented in a general way. I recommend that each of the aspects covered in the introduction should cover the topic of the study.

Response: The entire introduction section has been revised, according to your suggestion. However, we retained some of the information about the general population since there is a scarcity of studies that have analyzed the differences between sexes in these variables, specifically in this population. Thus, it seems essential to us to clarify what the literature says about the differences between men and women in the variables under analysis so that we can, subsequently, understand this topic in the population with disabilities.

Bibliographic index 2, present in lines 29-32, refers to another category of people, does not show the age of the subjects, it is from 1990, I recommend its deletion. Anthropometric trends have changed in the last 30 years, it is a secular trend.

Response: We removed the reference.

Subsection 2.3.5. I recommend mentioning the 6 dimensions of the questionnaire. I recommend mentioning the reliability coefficient for the entire questionnaire and for the 6 dimensions, value for each sample.

Response: Dear reviewer, we have mentioned the factors and dimensions of the questionnaire. We have also mentioned the internal consistency values for self-reports and third-party reports.

I recommend that caregivers be mentioned in the participants section as part of the experiment, or the clarification.

Response: Done.

Table 1, I recommend that the number of subjects be mentioned on the first or second line. Also under the table you mention p = significance level, it does not appear from the table, I recommend clarification.

Response: Dear reviewer, by mistake table 1 was not fully visible. We have added the number of participants.

Section 3 does not emerge from the table and its interpretation of the 6 dimensions of the Personal Outcomes Scale Portuguese version questionnaire, I recommend clarification. Only present a total value.

Response: As we mentioned earlier, due to a formatting mistake, part of the table was not visible. For this study, we only looked at the overall quality of life value.

I recommend that it be mentioned what is the proportion of sexes in the third party report, aspects related to numbers, gender, etc. about them were not mentioned in any section.

Response: Dear reviewer, we do not have access to this information about the third party report, since it was the professionals accredited for this purpose at the institution who implemented it for the families and we were not provided with this information.

Taking into account the statistical indicators mentioned in section 2.4, they do not fully emerge in section 3. Results. I recommend that all parameters calculated for each indicator should be mentioned in table 1. Mention what is the significance value p.

Response: Please see previous comment.

Section 2.4. I recommend that you mention that the reliability of the questionnaire and the indicator were calculated.

Response: Dear reviewer, the values are reported in section 2.3.5.

Lines 213-214 recommend mentioning which category of persons you refer to, this purpose is too general.

Response: Done.

Line 303 - I recommend the correction, those with profound deficiency are excluded from the study, according to the previously mentioned criteria. I recommend that it should be mentioned in the statements that no interventional program was applied.

Response: Thank you for your suggestion, which has been accepted.

I recommend rewriting the Conclusions section focused on the results of this study. I sound too general.

Response: The conclusion has been revised.

In conclusion, he recommends that the statistics be improved, concrete and relevant correlations be made, and the interpretation of the results be focused. The introduction should be majorly revised.

Response: Dear reviewer, thank you for your suggestions. We hope that the changes made will be in accordance with your comments and will substantially improve the article.

Reviewer 2 Report

Comments and Suggestions for Authors

Dear authors: In order to improve your manuscript, here are some suggestions that may be useful.

For the participants, I suggest that you separate them into at least one group with Intellectual Disabilities and another group with Developmental Disabilities. In that case, the results should distinguish between people with one or the other disability in both men and women. If possible, other tables could be introduced in the results indicating the variables studied in each disability group. This would lead to further discussion and conclusions. 

In addition, in order to identify the participants more precisely, you could indicate the specific illnesses they might have.

Also, you should specify the treatments, rehabilitation, physiotherapy, etc., that the participants undergo in order to determine the influence of these treatments on the results of the physical tests.

In the statistical analysis section, you use the Mann-Whitney U test, although a Student's t-test seems more appropriate (unless the distribution of the results was not normal or the homoscedasticity test was abnormal). Please clarify this aspect.

In the results, in table 1, please add a column with significance values. I think the Measures Total column can be deleted because it is not referred to in the rest of the manuscript.

In light of the results, and from an ethical point of view, please indicate whether there was feedback with the participants to establish physical activity or physiotherapy protocols to improve the deficits observed in the tasks assessed.

Best of luck to the authors.

Author Response

Response to REVIEWER 2

Dear authors: In order to improve your manuscript, here are some suggestions that may be useful.

For the participants, I suggest that you separate them into at least one group with Intellectual Disabilities and another group with Developmental Disabilities. In that case, the results should distinguish between people with one or the other disability in both men and women. If possible, other tables could be introduced in the results indicating the variables studied in each disability group. This would lead to further discussion and conclusions.

Response: Thank you for your comment. However, the diagnosis that the participants in this study have is of intellectual and developmental disability, considered in a unique and global dimension as reported in the literature. Intellectual and developmental disability (IDD) is a developmental disorder that originates during the individual’s developmental period, up to the age of 22. It is characterised by limitations in adaptive behaviour and intellectual functioning, which are expressed in the conceptual, practical and social domains with different degrees of severity.

  1. Schalock, R.L.; Luckasson, R.; Tassé, M.J. An Overview of Intellectual Disability: Definition, Diagnosis, Classification, and Systems of Supports (12th Ed.). Am. J. Intellect. Dev. Disabil. 2021, 126, 439–442.
  2. Rodrigues, A.R.; Santos, S.; Rodrigues, A.; Estevens, M.; Sousa, E. Executive Profile of Adults with Intellectual Disability and Psychomotor Intervention’ Effects on Executive Functioning. Physiother. Res. Rep. 2019, 2, 1–7.
  3. Cleaver, S.; Hunter, D.; Ouellette-Kuntz, H. Physical Mobility Limitations in Adults with Intellectual Disabilities: A Systematic Review. J. Intellect. Disabil. Res. 2009, 53, 93–105.

In addition, in order to identify the participants more precisely, you could indicate the specific illnesses they might have.

Response: As mentioned in the response to the previous comment, all participants were diagnosed with IDD. However, as stated in the participants section, individuals with severe IDD were excluded from the study.

Also, you should specify the treatments, rehabilitation, physiotherapy, etc., that the participants undergo in order to determine the influence of these treatments on the results of the physical tests.

Response: Dear reviewer, all the participants were institutionalized and carried out the usual activities of the institution, which are not structured. In addition, there was no control of diet or medication, among other variables that would allow us to make these causal relationships.

In the statistical analysis section, you use the Mann-Whitney U test, although a Student's t-test seems more appropriate (unless the distribution of the results was not normal or the homoscedasticity test was abnormal). Please clarify this aspect.

Response: Dear reviewer, we chose this statistical test because of the size of the sample.

In the results, in table 1, please add a column with significance values. I think the Measures Total column can be deleted because it is not referred to in the rest of the manuscript.

Response: Dear proofreader, due to a formatting mistake, part of the table was not visible. We have proceeded accordingly.

In light of the results, and from an ethical point of view, please indicate whether there was feedback with the participants to establish physical activity or physiotherapy protocols to improve the deficits observed in the tasks assessed.

Response: Feedback was given to the organization and the participants. In the same vein, this study gave rise to the development of an intervention project with the aim of providing regular practice for these participants.

Best of luck to the authors.

Round 2

Reviewer 1 Report

Comments and Suggestions for Authors

No comments

Author Response

Thank you for your comment.

Reviewer 2 Report

Comments and Suggestions for Authors

Dear authors, I believe that the changes made to the manuscript clarify and improve the paper overall. In the statistical analysis, precisely, if the number of participants is small, it is better to use the Student's t-test. In any case, to be clear, you should indicate whether or not the data were normal and, since you claim to have performed it, the homoscedasticity test.

Author Response

 We thank the reviewer for their comment. Indeed, our sample is less than 50 subjects. So, based on the recommendations of Ho (2014), particularly on the fact that we tend to be more conservative in terms of statistical analysis when the sample size is less than 50, which recommends the use of non-parametric techniques. However, information on the calculation of the normality test (Shapiro-Wilk) was included.

Ho, R. (2014). Handbook of Univariate and Multivariate Data Analysis with IBM SPSS. CRC Press.